# A Missense Variant in *SLC39A4* in a Litter of Turkish Van Cats with Acrodermatitis Enteropathica

**DOI:** 10.3390/genes12091309

**Published:** 2021-08-25

**Authors:** Sarah Kiener, Robert Cikota, Monika Welle, Vidhya Jagannathan, Susanne Åhman, Tosso Leeb

**Affiliations:** 1Institute of Genetics, Vetsuisse Faculty, University of Bern, 3001 Bern, Switzerland; sarah.kiener@vetsuisse.unibe.ch (S.K.); vidhya.jagannathan@vetsuisse.unibe.ch (V.J.); 2Dermfocus, University of Bern, 3001 Bern, Switzerland; monika.welle@vetsuisse.unibe.ch; 3VetaDerm Veterinärklinik, Järngatan 14, 234 35 Lomma, Sweden; robert@vetaderm.se (R.C.); susanne@vetaderm.se (S.Å.); 4Institute of Animal Pathology, Vetsuisse Faculty, University of Bern, 3001 Bern, Switzerland

**Keywords:** *Felis catus*, whole genome sequencing, dermatology, genodermatosis, zinc

## Abstract

In a litter of Turkish Van cats, three out of six kittens developed severe signs of skin disease, diarrhea, and systemic signs of stunted growth at 6 weeks of age. Massive secondary infections of the skin lesions evolved. Histopathological examinations showed a mild to moderate hyperplastic epidermis, covered by a thick layer of laminar to compact, mostly parakeratotic keratin. The dermis was infiltrated with moderate amounts of lymphocytes and plasma cells. Due to the severity of the clinical signs, one affected kitten died and the other two had to be euthanized. We sequenced the genome of one affected kitten and compared the data to 54 control genomes. A search for private variants in the two candidate genes for the observed phenotype, *MKLN1* and *SLC39A4*, revealed a single protein-changing variant, *SLC39A4*:c.1057G>C or p.Gly353Arg. The solute carrier family 39 member 4 gene (*SLC39A4*) encodes an intestinal zinc transporter required for the uptake of dietary zinc. The variant is predicted to change a highly conserved glycine residue within the first transmembrane domain, which most likely leads to a loss of function. The genotypes of the index family showed the expected co-segregation with the phenotype and the mutant allele was absent from 173 unrelated control cats. Together with the knowledge on the effects of *SLC39A4* variants in other species, these data suggest *SLC39A4*:c.1057G>C as candidate causative genetic variant for the phenotype in the investigated kittens. In line with the human phenotype, we propose to designate this disease acrodermatitis enteropathica (AE).

## 1. Introduction

Lethal acrodermatitis (LAD) is a monogenic autosomal recessive disease in Bull Terriers and Miniature Bull Terriers (OMIA 002146-9615). It is characterized by skin lesions on the feet and face, diarrhea, bronchopneumonia, and a failure to thrive [1,2,3]. Some studies also observed a decreased plasma zinc level in LAD-affected dogs [2,4]. Oral or parenteral supplementation of zinc, however, did not improve the clinical signs of these dogs [1]. LAD in dogs is caused by a splice defect in the *MKLN1* gene [5]. This gene encodes the muskelin 1 protein, which is, intracellularly, widely expressed and is discussed to be involved in several functions, including cell adhesion, morphology, spreading, and intracellular transport processes [6,7,8,9,10,11,12,13,14,15,16]. To date, the exact pathogenesis of LAD has not been elucidated [5].

Acrodermatitis enteropathica (AE) is a related phenotype, which is also inherited as an autosomal recessive trait. It has been described in humans (OMIM # 201100) and cattle (OMIA 000593-9913) [17,18]. The bovine disease was termed lethal trait A46, bovine hereditary zinc deficiency or Adema disease. Clinical findings were rapidly associated with low plasma zinc levels [18,19,20,21,22].

Zinc ions (Zn^2+^) are essential for many biological processes. More than 300 enzymes need zinc as an essential co-factor, and approximately 2800 human proteins are potentially zinc-binding. In these metalloproteins, the binding of zinc is needed either because the metal ion is involved in the catalytic mechanism, or because it stabilizes the protein tertiary or quaternary structure [23,24]. In zinc-deficient individuals, the clinical manifestations are correspondingly diverse.

AE in humans and cattle is caused by variants in the *SCL39A4* gene, encoding the solute carrier family 39 member 4, an intestinal zinc transporter that has also been termed ZIP4 [25,26,27,28]. The loss of function of this transporter leads to a severe systemic zinc deficiency. Untreated, the disease ends fatally. However, as there is another, non-saturable transport pathway besides the high-affinity SLC39A4 transporter, the zinc deficit in AE patients can be successfully treated with sufficient supplementation of oral zinc [18,19,21,28,29,30,31,32,33,34].

In the present study, we investigated a litter of Turkish Van cats with skin lesions and additional clinical signs of zinc deficiency. The goal of the study was to characterize the clinical and histopathological phenotype and to identify the underlying causative genetic variant.

## 2. Materials and Methods

### 2.1. Animal Selection

This study included a Turkish Van cat family with 12 animals and 173 genetically diverse control cats from other breeds. Genomic DNA was isolated from ETDA blood samples with the Maxwell RSC Whole Blood Kit using a Maxwell RSC instrument (Promega, Dübendorf, Switzerland).

### 2.2. Histopathological Examinations

Skin biopsies (6 mm punch and wedge incision) were taken under general anesthesia from Case 1 and 2 at the time of presentation at 10 weeks of age, when Case 1 was euthanized. Further biopsies were taken from Case 3 at the time of euthanasia, two weeks after the initial presentation, at 12 weeks of age. Areas biopsied included the paw pads, claw folds, axilla, inguinal region, and concave pinnae. From the euthanized kittens, tissue samples were also taken from muco-cutaneous junctions, the thoracic esophagus, stomach, and small intestine. The samples were fixed in 10% neutral buffered formalin and routinely processed including staining with hematoxylin and eosin (HE) and periodic acid Schiff (PAS).

### 2.3. Whole Genome Sequencing

An Illumina TruSeq PCR-free DNA library with ~400 bp insert size of an AE affected Turkish Van cat was prepared. We collected 188 million 2 × 150 bp paired-end reads or 20.7× coverage on a NovaSeq 6000 instrument. The reads were mapped to the FelCat9.0 cat reference genome assembly and aligned as described [35]. The sequence data were submitted to the European Nucleotide Archive with the study accession PRJEB7401 and sample accession SAMEA8609184.

### 2.4. Variant Calling

Variant calling was performed as described [35]. The SnpEff software was used to predict the functional effects of the called variants [36] together with NCBI annotation release 104 for the FelCat9.0 genome reference assembly. For variant filtering, we used 54 control genomes (Appendix A). The control genomes were derived from 29 purebred cats of 8 different breeds and 25 random-bred domestic cats.

### 2.5. Gene Analysis

Numbering within the feline *SLC39A4* gene corresponds to the NCBI RefSeq accession numbers XM_004000173.3 (mRNA) and XP_004000222.2 (protein). We performed a multiple species comparison of orthologous SLC39A4 amino acid sequences with the accessions NP_570901.2 (*Homo sapiens*), XP_001157597.3 (*Pan troglodytes*), XP_001098635.2 (*Macaca mulatta*), XP_005628372.1 (*Canis lupus familiaris*), NP_001039532.1 (*Bos taurus*), NP_082340.1 (*Mus musculus*), and NP_001071137.1 (*Rattus norvegicus*). Protein alignments were either directly taken from the precomputed NCBI HomoloGene database [37] or performed with blastp on the NCBI BLAST server [38].

### 2.6. Sanger Sequencing

Sanger sequencing of PCR amplicons was used to confirm the candidate variant *SCL39A4*:c.1057G>C and to genotype cats. With the primers 5′-GGA TGG GGC TTT AAG GGT TA-3′ (Primer F) and 5′-GCT GAC CTT GGG TGT CAA GT-3′ (Primer R) a 334 bp PCR product was amplified from genomic DNA using AmpliTaqGold360Mastermix (Thermo Fisher Scientific, Waltham, MA, USA). After treatment with shrimp alkaline phosphatase and exonuclease I, PCR amplicons were sequenced on an ABI 3730 DNA Analyzer (Thermo Fisher Scientific). The Sequencher 5.1 software was used to analyze the Sanger sequences (GeneCodes, Ann Arbor, MI, USA).

## 3. Results

### 3.1. Family Anamnesis, Clinical Examinations and Histopathology

A 10 week-old Turkish Van litter was referred due to a 4 week history of rapidly progressing, severe dermatological and systemic disease in three out of six kittens. The phenotype distribution was suggestive of a monogenic autosomal recessive mode of inheritance (Figure 1).

All kittens had a normal development and growth until 6 weeks of age. The mother was healthy. Starting from 6–8 weeks of age, the three affected kittens displayed similar systemic and dermatological signs of varying severity. The affected cats showed growth retardation and developed diarrhea. Dermatological signs included widespread and severe scaling, alopecia, and moist dermatitis as well as numerous erosions and focal ulcerations. Lesions had a predominantly ventral and distal distribution, most severely in friction areas but also around mucocutaneous junctions, concave pinnae and paw pads (Figure 2). Furthermore, there was a widespread and severe secondary pyoderma. Interdigital crusts and erosions were present in clawfolds, but the claws were normal. Several paw pads had splitting lesions in a horizontal direction across the complete pad and erosions or ulcerations in the junction between paw pad and skin (Figure 2d). There were no visible oral lesions and teeth were normal. Systemic signs included anorexia, fever, dehydration, decreased body weight (on average 40% less than unaffected siblings). Thoracic auscultation, abdominal and lymph node palpation were all unremarkable. Prior to referral, the affected kittens had tested negative for Demodex gatoi and for dermatophytosis via a PCR test. Clinical signs had not improved upon treatment with systemic amoxicillin and prednisolone for 5 days. Complete blood counts were unremarkable.

The histological findings were similar in all samples from the three kittens. The epidermis was mildly to moderately hyperplastic, and the granular cell layer was thin to absent. Few apoptotic cells were present in the superficial epidermis in some samples. The epidermis was diffusely covered by variable amounts of laminar to compact, parakeratotic keratin, characterized by retained nuclei in the stratum corneum. In between the corneocytes, numerous cocci and/or yeasts were present. Multifocally, the biopsies were covered by large serocellular crusts, composed of corneocytes, proteinaceous material, degenerate inflammatory cells, and abundant cocci. In the epidermis, underneath some of the crusts, spongiosis and leukocyte exocytosis was present. In the superficial dermis, a mild perivascular infiltrate composed of mast cells, few lymphocytes and rare plasma cells was noted. In the biopsies of one kitten, there were also some eosinophils. Multifocally, the epidermis was ulcerated and there were abundant inflammatory cells underneath the ulcer (Figure 3). Findings in non-haired skin from the paw pads were similar (Figure 3b).

Two affected kittens were euthanized due to rapidly progressing and severe generalized disease at 10 (Case 1) and 12 (Case 3) weeks of age. The third affected kitten, Case 2, appeared relatively bright, playful and gained weight, until 12 weeks of age, when it suddenly and rapidly deteriorated and died before it could be euthanized. The three unaffected siblings have continued to grow and develop into healthy adult individuals.

### 3.2. Genetic Analysis

In order to characterize the underlying causative genetic variant, we sequenced the genome of Case 2 at 20.7× coverage and searched for homozygous variants in *MKLN1* and *SLC39A4*, the two candidate genes for the observed phenotype. We filtered for private variants, which were exclusively present in the affected cat and absent from the genomes of 54 other cats (Table 1 and Appendix A).

This analysis identified a single homozygous private protein-changing candidate variant in *SLC39A4*. The variant can be designated as ChrF2:85,320,523C>G (FelCat9.0). It was a missense variant, XM_004000173.3:c.1057G>C, predicted to change a highly conserved glycine residue in the first transmembrane domain of the zinc transporter, XP_004000222.2:p.(Gly353Arg).

We confirmed the presence of the *SLC39A4* missense variant by Sanger sequencing (Figure 4). Genotypes at the variant perfectly co-segregated with the phenotype in the available family members. All three affected cats were homozygous. The parents and two close relatives carried the mutant allele in a heterozygous state. All other non-affected cats were homozygous for the wildtype allele (Figure 1).

We determined the genotypes at *SLC39A4*:c.1057G>C for the index family as well as 173 other cats. The mutant allele was exclusively detected in the Turkish Van family and not in cats from other breeds. All non-affected cats were either heterozygous or homozygous for the wildtype allele (Table 2).

## 4. Discussion

The characteristic clinical signs and the early age of onset in the three affected cats described in this study strongly suggested the presence of an inherited disease. The clinical phenotype mostly resembled that of lethal acrodermatosis (LAD), previously described in Bull Terriers and Miniature Bull Terriers, and acrodermatitis enteropathica (AE), which is known in humans and was also described in cattle under the name lethal trait A46, bovine hereditary zinc deficiency, or Adema disease [1,2,3,17,18]. The histopathological findings are unique and have not been reported in cats before. They strikingly resembled the findings seen in lethal acrodermatitis in Bull terriers and miniature Bull terriers [1,2,3,4,5] or other disorders associated with impaired zinc absorption, such as canine zinc responsive dermatosis [40,41] or bovine hereditary zinc deficiency [17,18]. Another disorder with diffuse parakeratosis described in dogs and very rarely in cats is superficial necrolytic dermatosis; however, this disorder is associated with either hepatic disease or hepatic or pancreatic neoplasia and seen in much older individuals. Thallium toxicosis also presents with severe parakeratotic hyperkeratosis but is very unlikely in such young kittens and is nowadays rare, since thallium was banned as rodenticide [40,41]. A last differential would have been a generalized form of non-epidermolytic ichthyosis with secondary infection. However, in generalized ichthyosis, the stratum corneum is usually orthokeratotic [40,41].

The genome of one affected cat was sequenced and filtered for private variants in the two candidate genes for both differential diagnoses. The analysis revealed a private missense variant in the *SLC39A4* gene encoding the solute carrier family 39 member 4, a zinc transporter with eight transmembrane domains, which is responsible for the saturable zinc uptake in the small intestine [29,39]. Loss of function of SLC39A4 leads to a severe zinc deficiency, which ends fatally if untreated. The fact that there is another, unsaturable intestinal zinc transport pathway enables the treatment of the deficiency with oral supplementation of large doses of zinc [18,19,21,28,29,30,31,32,33,34].

Missense variants in transmembrane domains were previously described to impair the function of SLC39A4 [39]. So far, one variant in the first transmembrane domain has been reported in a human patient [26]. The identified missense variant in the affected cats changes a highly conserved glycine residue in the first transmembrane domain. We assume that the change in the uncharged glycine to the positively charged arginine is intolerable for the integration of the protein in the membrane, thus causing a complete loss of function of the transporter. This hypothesis is further supported by the fact that four of the known pathogenic human variants involve an exchange of a glycine within a transmembrane domain to a charged amino acid.

We have to caution that we did not experimentally confirm the loss of function of SLC39A4 on the protein level. While the comparative data from human and cattle clearly suggest that *SLC39A4*:c.1057G>C is a very likely candidate causative variant for the observed skin lesions in the affected cats, further experiments are required to obtain definitive proof for this hypothesis.

## 5. Conclusions

In this study, we provide the first report of acrodermatitis enteropathica in cats. The identification of a candidate causative variant, *SLC39A4*:c.1057G>C, enables genetic testing to prevent future breeding of affected cats. For affected kittens, the genetic test also offers the chance to initiate adequate therapy, since AE is treatable with oral zinc supplementation.

## Figures and Tables

**Figure 1 genes-12-01309-f001:**
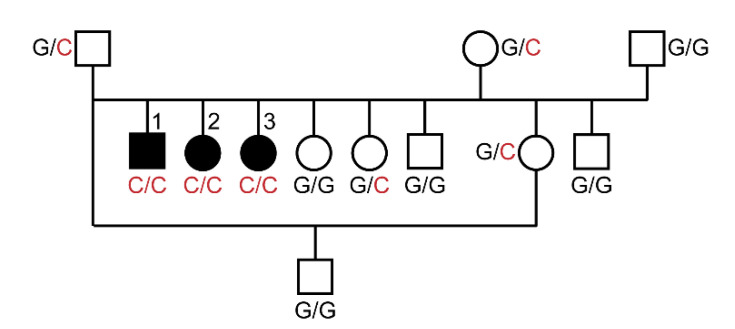
Pedigree of the investigated Turkish Van family. Squares represent males and circles females. The three affected kittens are numbered and indicated by filled symbols. Genotypes at the *SCL39A4*:c.1057G>C variant are indicated.

**Figure 2 genes-12-01309-f002:**
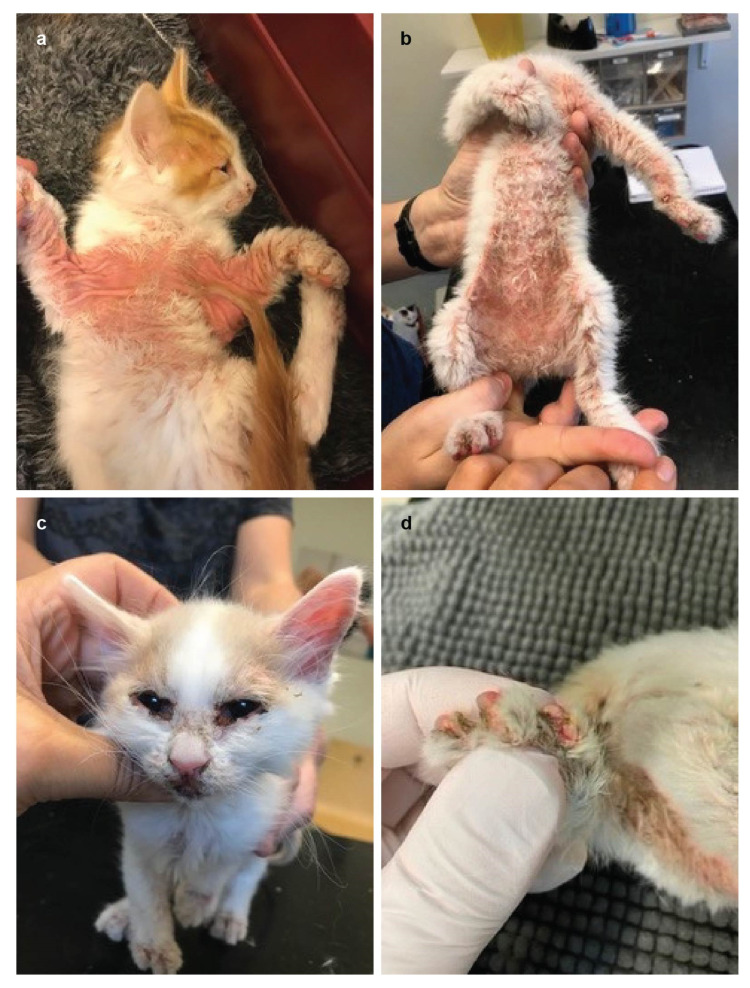
Clinical phenotype of cats affected with AE. (**a**) Skin disease with erythema and beginning alopecia on the ventrum. (**b**) More severe presentation with erythema, crusting, scaling, and alopecia on the ventral and distal aspects. (**c**) Characteristic skin lesions in the face and the ear pinnae. (**d**) Interdigital erosions.

**Figure 3 genes-12-01309-f003:**
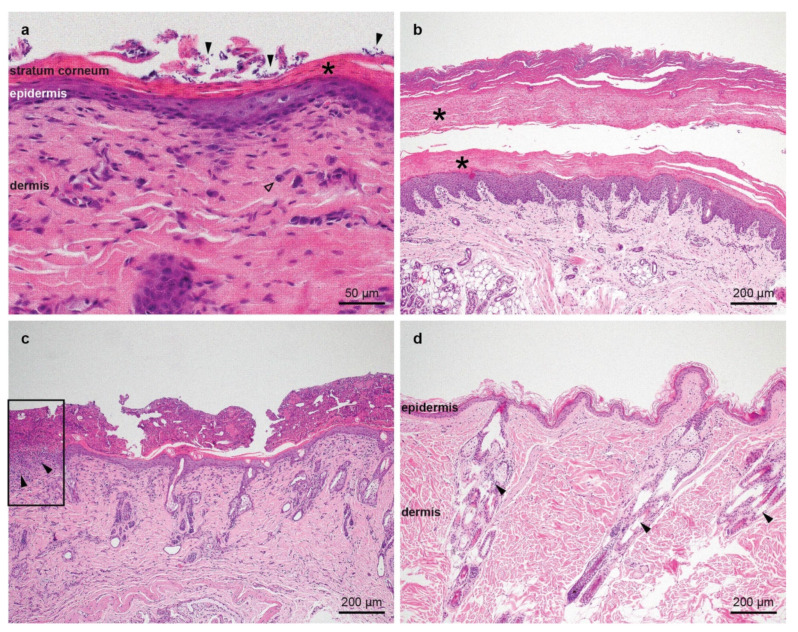
Histopathological images of representative skin biopsies, hematoxillin and eosin stain. (**a**) The epidermis is mildly hyperplastic and covered by compact parakeratotic keratin, characterized by the presence of retained nuclei in the stratum corneum (asterisk). There are numerous yeasts in between the corneocytes, indicated by the filled arrows. There is a mild perivascular infiltrate in the superficial dermis composed mostly of mast cells (empty arrow). (**b**) The epidermis of the paw pad is mildly hyperplastic and covered by a thick layer of compact parakeratotic keratin (asterisks). (**c**) The epidermis is mildly hyperplastic and covered by a thin layer of compact parakeratotic keratin. The stratum corneum is covered by a thick serocellular crust, composed of corneocytes, proteinaceous material, degenerate inflammatory cells, and abundant cocci. In the rectangle on the left edge of the image, an ulcer is present, characterized by the lack of the epidermis and abundant inflammatory cells in the superficial dermis underneath the ulcer (filled arrows). (**d**) Skin of a healthy cat as control showing the epidermis of normal thickness covered by basketweave orthokeratotic keratin together with the dermis and the adnexa (filled arrows).

**Figure 4 genes-12-01309-f004:**
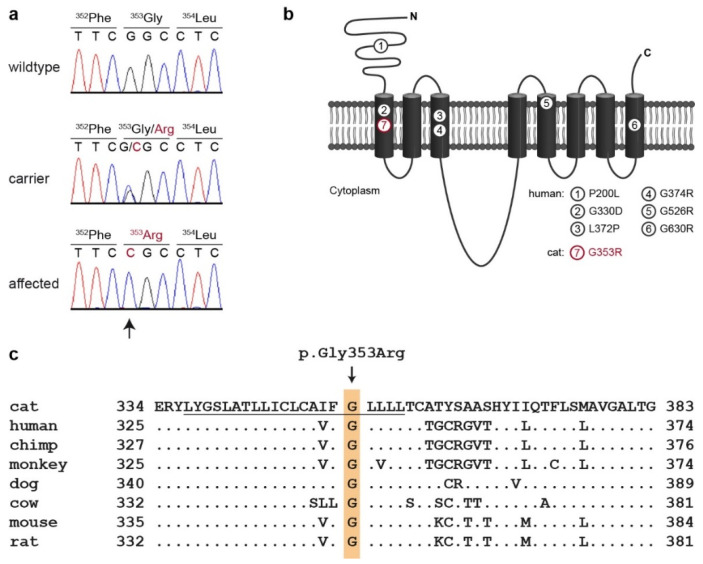
Details of the *SLC39A4*:c.1057G>C, p.Gly353Arg variant. (**a**) Representative electropherograms of three cats with different genotypes are shown. The variable position is indicated by an arrow and the amino acid translations are shown. (**b**) Model of SLC39A4 membrane topology, modified from [39]. The positions of six functionally characterized missense variants identified in human AE patients are indicated. The position of the feline p.Gly353Arg variant identified in the investigated cat is indicated in red. (**c**) Multiple-species alignment of the SLC39A4 amino acid sequences in the region harboring the p.Gly353Arg variant. The variant affects a highly conserved glycine residue. The sequence of the first transmembrane domain is underlined.

**Table 1 genes-12-01309-t001:** Results of variant filtering in the affected Turkish Van cat against 54 control genomes. Only homozygous variants are reported.

Filtering Step	Variants
all variants in the affected cat	5,518,410
private variants	56,054
protein-changing private variants	161
protein-changing private variants in *MKLN1* or *SLC39A4*	1

**Table 2 genes-12-01309-t002:** Genotype–phenotype association of the *SCL39A4*:c.1057G>C variant with AE.

Cats	G/G	G/C	C/C
Cases (*n* = 3)	-	-	3
Controls, Turkish Van cats from index family (*n* = 9)	5	4	-
Controls, other breeds (*n* = 173)	173	-	-

## Data Availability

The genome sequence data were submitted to the European Nucleotide Archive (ENA). All accession numbers are listed in Appendix A.

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
