# Peer review of "A Missense Variant in SLC39A4 in a Litter of Turkish Van Cats with Acrodermatitis Enteropathica"

_genes, 2021, doi:10.3390/genes12091309_

Round 1

Reviewer 1 Report

I would like to thank the authors for this manuscript of high quality and high interest.

My only suggestion, at line 209, is to add other causes of SND like: pancreatic glucagon-producing alpha cell tumor, insulin-producing pancreatic islet cell carcinoma.

Author Response

(1)

My only suggestion, at line 209, is to add other causes of SND like: pancreatic glucagon-producing alpha cell tumor, insulin-producing pancreatic islet cell carcinoma.

Response: We added these potential differentials accordingly.

Reviewer 2 Report

The authors describe an acrodermatis enteropathica in three Turkish Van cats from one litter and the identification of a mutation in SLC39A4 gene that may be implicated in the disease.

The paper is well written and easy to follow, but I have some comments and questions.

In the introduction section:

- As several species are cited throughout the paper, the species should be specified: “In dog breeds, LAD is caused by a splice defect …” line 38; “In humans, AE is caused by variants in the SLC39A4 gene …” line 54.

In the materials and methods section:

- line 73, I do not understand when the skin biopsies were realized for the 2nd and 3rd kitten: it is said first at the time of euthanasia, and then at presentation and two weeks later in parenthesis. This is confusing.

- line 90, the authors should precise the number of cat breeds that make the 54 control genomes.

- line 95, the authors should precise the program they used for the multiple species alignment.

In the results section:

- Figure 1: the individuals should be numbered, so the text can refer to this numbering. So, line 167, the cat for which the whole genome was sequenced can be indicated. A word or two about why this cat was chosen for the whole genome sequencing will be nice.

- Figure 2, histopathological images e to g: there is no annotation on the images. I do not think that people that will read the paper will be familiar with this type of images so they should be annotated to show the different skin layers: epidermis, layer of compact parakeratotic keratin, stratum corneum. It is said in the legend (line 156) as well as in the text (lines 140 and 142) that cocci and/or yeasts can be observed. How do you recognize them? One or two arrows showing them on the figure will be nice. Also the three images are from affected cats. One image of an unaffected cat is needed so the readers can also appreciate the difference with a “normal” situation. In the g image, a square or something else should delimit the ulcer, and an arrow should point to the abundant inflammatory cells. Finally, the images should have a scale bar (it seems that f and g were taken at a higher magnification compared to e).

- Table 1 and following text: only two genes were looked because they are known in humans and dogs. But what about the genes containing the 160 other variants? How many genes? Any interesting one? Did you look at them to exclude them? It is important to precise as only the variant in the SLC39A4 gene was checked in the index family and in the 173 other cats. Even if the variant seems to fit perfectly in term of segregation and phenotype, other relevant variant could be found in the remaining 160, unless all are excluded.

- Table S2 concerning the private variants of the sequenced cat. As there are 157 807 lines in the table, it is quite hard to find the LAMB3 variant, even if it is highlighted in yellow, line 153 802… I suggest to make a third excel file, which will only contains the 161 protein-changing private variants. Also as one variant can be counted several times, it would be nice to give each variant a number (so we end up at 161). Do you count the 5’ UTR premature start codon as protein changing variants and are these variants inframe?

- Figure 3, c: alignment of the proteins. It seems that for this part of the protein all species have the same number of amino acids. So I do not understand why the cat sequence starts at 334 and the human one at 325 if they both end at 374… (same with rat). I think there is a problem on the numbering.

- line 193: It is only in table 2 that we know that in the 173 are “genetically diverse” cats (lines 66-67), there is no Turkish Van cats. This should already be mentioned lines 66-67 in the materials and methods section. I understand that the Turkish Van cats controls are all part of the family presented in figure 1. They should not be considered as controls as they were used to show the segregation of the variant in the family.

In the discussion section

- any way to measure if the kittens had low intake of zinc?

- again one or two sentences on the other genes in which homozygous protein changing variants were found in the whole genome sequence of the affected cat: are they all irrelevant?

- as the authors have skin biopsies, they should perform immunohistochemistry to assess the hypothesis that the variant will impair the integration of the protein in the plasma membrane (line 228). This will greatly reinforce the implication of the found variant.

Author Response

(1)

In the introduction section:

As several species are cited throughout the paper, the species should be specified: “In dog breeds, LAD is caused by a splice defect …” line 38; “In humans, AE is caused by variants in the SLC39A4 gene …” line 54.

Response: We added the species names at the two indicated instances.

(2)

In the materials and methods section:

line 73, I do not understand when the skin biopsies were realized for the 2nd and 3rd kitten: it is said first at the time of euthanasia, and then at presentation and two weeks later in parenthesis. This is confusing.

Response: We revised the text and hope that it is now easier to understand.

(3)

line 90, the authors should precise the number of cat breeds that make the 54 control genomes.

Response: We added the breed information as requested.

(4)

line 95, the authors should precise the program they used for the multiple species alignment.

Response: We added the requested information.

(5)

In the results section:

Figure 1: the individuals should be numbered, so the text can refer to this numbering. So, line 167, the cat for which the whole genome was sequenced can be indicated. A word or two about why this cat was chosen for the whole genome sequencing will be nice.

Response: We numbered the three cases as requested. We chose a female cat for sequencing as this gives better coverage on the X-chromosome, which might be important if we use this genome as control in future studies. As this rationale is not directly related to the present study, we did not elaborate on this in the manuscript.

(6)

Figure 2, histopathological images e to g: there is no annotation on the images. I do not think that people that will read the paper will be familiar with this type of images so they should be annotated to show the different skin layers: epidermis, layer of compact parakeratotic keratin, stratum corneum. It is said in the legend (line 156) as well as in the text (lines 140 and 142) that cocci and/or yeasts can be observed. How do you recognize them? One or two arrows showing them on the figure will be nice. Also the three images are from affected cats. One image of an unaffected cat is needed so the readers can also appreciate the difference with a “normal” situation. In the g image, a square or something else should delimit the ulcer, and an arrow should point to the abundant inflammatory cells. Finally, the images should have a scale bar (it seems that f and g were taken at a higher magnification compared to e).

Response: We completely revised the figure and split it into two separate figures for the clinical photos (Figure 2) and the histopathology (Figure 3). The histopathology micrographs are now bigger, which makes it easier to see the changes. We labeled important structures and explained them in the legend. We also added a micrograph from control skin at the same resolution (Figure 3d).

(7)

Table S2 concerning the private variants of the sequenced cat. As there are 157 807 lines in the table, it is quite hard to find the LAMB3 variant, even if it is highlighted in yellow, line 153 802… I suggest to make a third excel file, which will only contains the 161 protein-changing private variants. Also as one variant can be counted several times, it would be nice to give each variant a number (so we end up at 161). Do you count the 5’ UTR premature start codon as protein changing variants and are these variants inframe?

Response: We added another sheet to Table S2 that contains the 161 protein changing variants. In the header to this table, we state our definition of “protein changing”, which only includes variants with SnpEff impact prediction of “high” or “moderate”. SnpEff classifies the gain of an additional start codon in the 5’-UTR as low impact and such variants are not contained in the 161 protein changing variants.

(8)

Figure 3, c: alignment of the proteins. It seems that for this part of the protein all species have the same number of amino acids. So I do not understand why the cat sequence starts at 334 and the human one at 325 if they both end at 374… (same with rat). I think there is a problem on the numbering.

Response: We thank the reviewer very much for spotting the errors in the cat and rat amino acid numbering. The two errors were corrected accordingly.

(9)

line 193: It is only in table 2 that we know that in the 173 are “genetically diverse” cats (lines 66-67), there is no Turkish Van cats. This should already be mentioned lines 66-67 in the materials and methods section. I understand that the Turkish Van cats controls are all part of the family presented in figure 1. They should not be considered as controls as they were used to show the segregation of the variant in the family.

Response: We specified in the Material & Methods section that the 173 controls cats did not include any Turkish Van cats. We revised Table 2, so that it becomes clear that the 9 Turkish Van controls are the same animals as the ones shown in Figure 1. We suggest to keep these animals in the table to have a comprehensive overview of all genotypes at SCL39A4:c.1057G>C that were obtained in this study.

(10)

In the discussion section:

Any way to measure if the kittens had low intake of zinc?

Response: Unfortunately, we do not have this information. Please keep in mind that the three affected cats were privately owned and died or were euthanized at an early age. Only several months after the death of these kittens, the genetic analysis revealed that the underlying defect most likely was due to a defective zinc transporter.

(11)

Again one or two sentences on the other genes in which homozygous protein changing variants were found in the whole genome sequence of the affected cat: are they all irrelevant?

Response: We appreciate this comment. While some of the other protein changing variants can be excluded as being causative for the observed skin lesions based on the known functions of the affected proteins, there are also a large number of protein-changing variants that affect proteins whose in vivo function is incompletely or not at all characterized. We hope that the wording in our manuscript is sufficiently clear and cautious to indicate that we do not claim to have definitively proven the causality of the identified variant. E.g. lines 26/27: “Together with the knowledge on the effects of SLC39A4 variants in other species, these data suggest SLC39A4:c.1057G>C as candidate causative genetic variant for the phenotype in the investigated kittens.

We added another short paragraph at the end of the discussion emphasizing the limitations of our study. We don’t think it is useful to the reader, if we start speculations on hundreds of protein changing variants and tens of thousands of other genetic variants whose functional significance is mostly unclear at this moment.

(12)

As the authors have skin biopsies, they should perform immunohistochemistry to assess the hypothesis that the variant will impair the integration of the protein in the plasma membrane (line 228). This will greatly reinforce the implication of the found variant.

Response: We agree with the reviewer that an experimentally confirmed change on the protein level would greatly strengthen the claimed pathogenicity of the variant. Unfortunately, IHC for the feline SLC39A4 is not established. Commercial antibodies against human or murine proteins do not always work well in cat samples and such an IHC experiment would require some optimization. We currently do not have the resources to perform this experiment and hope that the manuscript may be published without this additional experiment.